# APC: Predict Global Representation From Local Observation In Multi-Agent Reinforcement Learning

## Abstract

Multi-agent reinforcement learning (MARL) algorithms with sequential decision-making strategies have achieved great success in cooperation tasks recently. To overcome the non-stationarity problem, these methods design a centralized controller that takes global observation as input and chooses actions for each agent in sequence. However, in most scenarios, global information is only available at training time, while agents act synchronously with their local observation at execution time, which prevents agents from leveraging more information in cooperation. In this paper, based on actor-critic architecture, we propose the actor-predicts-critic (APC) algorithm, in which the actor learns to predict the global representations of centralized critic from local observation. During the training, the actor not only receives the estimated state values, but also takes the critic's representations that are extracted from global information as the prediction targets. Since these global representations are closely related to agents' goals and rewards, agents can achieve better cooperation on MARL tasks utilizing the predicted representations. To prove the validity of APC, we evaluate the algorithm on StarCraftII, Google Research Football, and Multi-Agent Mujoco benchmarks. The results show that APC significantly outperforms the strong baselines in centralized training and decentralized execution (CTDE) framework, including MAT-Dec, MAPPO, and fine-tuned QMIX.

## 1 Introduction

Multi-agent reinforcement learning (MARL) (Buşoniu et al., 2010) over cooperation tasks aims to train a group of agents to maximize the rewards by cooperating with each other. The algorithms in MARL have displayed outstanding performance in various tasks and games like Dota 2 (Berner et al., 2019) and StarCraftII (Vinyals et al., 2019). Early MARL algorithms generally include two approaches: centralized learning and decentralized learning. Centralized learning (Claus & Boutilier, 1998; Guestrin et al., 2001) uses a centralized controller to make decisions for all the agents, which leads to the exponential explosion problem in the joint action space. Decentralized learning (Tampuu et al., 2017) makes agents train and execute separately, which leads to the non-stationarity problem and the algorithm may not converge. To combine the advantages of both methods and avoid suffering from their problems, researchers propose the *centralized training and decentralized execution (CTDE)* (Foerster et al., 2018), where agents are trained with global information and act separately with their local observation in execution. This framework provides convenience for generalizing algorithms from single-agent reinforcement learning to MARL, and these methods have made significant progress in various benchmarks. However, recent studies assert that algorithms in the CTDE framework still suffer from the non-stationarity problem. These studies, including MAT (Wen et al., 2022) and ACE (Li et al., 2022), propose sequential decision-making methods that arrange the agents in a certain order and use a centralized controller to make decisions for agents sequentially based on global observation and preceding agents' actions. With the global information and action dependency, these methods obtain excellent performance and beat the strong baselines in the CTDE framework. Moreover, the exponential explosion problem of centralized controllers is solved through the sequential search policy in joint action space.

However, in most scenarios, global states are only available at training time, and agents have to act synchronously and only have access to the local observation at execution time. Therefore, the application scenarios of the sequential decision-making methods which heavily rely on the global information and action-dependency are limited compared to CTDE methods. But, if we can combine their idea with the CTDE framework and allow agents to benefit from global information and action-dependency in the decentralized execution phase, they should not only achieve better performance but also have various application scenarios. A simple idea is to make agents predict global information and other agents' actions. However, since agents in the CTDE framework act synchronously, it is self-contradictory and meaningless to estimate other agents' actions and make decisions based on the estimation (see Appendix A). Fortunately, predicting global information with local observation is possible since we can use historical observations to bridge the gap between local and global information. Similar ideas have already appeared in agent-modeling MARL algorithms where agents learn to predict other agents' information. These methods usually utilize an encoder-decoder structure. Agents learn to encode observations into representations and the decoder decodes these representations into other agents' information to explain the validity of these representations.

In fact, in actor-critic architecture (Sutton et al., 1999; Lowe et al., 2017), there is already such an encoder-decoder structure existing in the critic network. The centralized critic encodes the global information into global representations, and then uses the global representations to estimate the state-action values of agents. Since these representations are used to estimate state-action values, we believe that they embed the information most directly relevant to agents' goals from the critic's global input. Therefore, instead of directly predicting global states/observations, we can predict the global representations of the centralized critic, which contains structured information that has been pre-processed. In addition, since the convergence of the critic network is faster and more stable compared to the actor network, learning the intermediate representations of the critic network will increase the stability of the actor network in training.

In this work, we aim to make agents learn to predict global representations with local observations. Based on the actor-critic architecture in the CTDE framework, we propose the actor-predicts-critic (APC) algorithm. In APC, the actor utilizes current and historical local observations to predict the centralized critic's global representations. The predicted representations will help agents choose optimal actions and cooperate with others. To demonstrate the validity of APC, we evaluate it with strong baselines on various benchmarks about cooperation tasks including StarCraft Multi-Agent Challenge (SMAC) (Samvelyan et al., 2019), Google Research Football (GRF) (Kurach et al., 2020) and Multi-Agent Mujoco (Peng et al., 2021). The results show that APC significantly outperforms other baselines in the CTDE framework, especially in hard tasks where only local observation is not enough to achieve effective cooperation. The ablation study also proves the effectiveness of predicting global representations for agents in the decision-making process.

## 2 RELATED WORKS

The MARL algorithms in the CTDE framework are divided into two categories. The first one is value-decomposition (VD) methods (Sunehag et al., 2017; Rashid et al., 2020b). Based on the IGM principle, these methods aim to find a reasonable way to assign the centralized state-action value to each agent. As the IGM principle requires, the assignment of the centralized value has to guarantee that the optimal actions of each agent based on their individual state-action values should consist of the optimal joint action based on centralized state-action values. VDN (Sunehag et al., 2017) distributes the centralized state-action value function equally to each agent. However, this simple division makes the centralized value function only a sum of individual values and takes no advantage of the global states in training. To provide a much richer class of state-action value function, QMIX (Rashid et al., 2020b) utilizes an additional mixing network to mix the individual values into the centralized value, while the parameters of the mixing network are set positive to guarantee the IGM principle. Inspired by QMIX, various studies (Son et al., 2019; Rashid et al., 2020a; Wang et al., 2021) concentrate on optimizing the centralized Q-value assignment following the IGM principle or directly optimizing the IGM principle. Recent study (Hu et al., 2021) shows that after being fine-tuned with proper tricks, the results of QMIX obtain a significant improvement on SMAC tasks, which reveals the potential of value-based algorithms. Although these methods have performed impressive results on several benchmarks, they can only be applied to tasks with discrete action space due to the use of the state-action value function.

To solve MARL problems with continuous action space, another kind of CTDE method utilizes policy gradient (PG) algorithms and extends it from reinforcement learning (RL) to MARL. MAD-DPG (Lowe et al., 2017) uses the actor-critic structure in the CTDE framework to improve the performance of the PG algorithm in MARL. Based on this, COMA (Foerster et al., 2018) uses the counterfactual baseline to handle the challenge of multi-agent credit assignment. That means the algorithm will marginalize a single agent's action and keep the actions of other agents fixed to figure out the contribution of each agent to the overall reward. IPPO (de Witt et al., 2020) and MAPPO (Yu et al., 2022) generalize the trust region learning from RL to MARL. IPPO proves that the PPO algorithm works excellently in MARL even if the agents are trained independently. With CTDE architecture, MAPPO demonstrates that PPO outperforms most baselines in MARL scenarios. These methods based on PG have achieved great success in MARL tasks with both discrete and continuous action spaces. Most of these approaches follow the actor-critic architecture (Lowe et al., 2017; Foerster et al., 2018), where a centralized critic takes global information as input and the actor takes local observations as input. The actor only learns from the state-action values assigned by the critic and cannot benefit directly from the global information.

With the development of CTDE methods, some researchers have discovered an unavoidable disadvantage of decentralized execution. They point out that CTDE methods still suffer from the non-stationarity problem, as most of them utilize shared parameters between agents and make decisions simultaneously. In that case, agents' local optimization of their own actions can jointly lead to a worse outcome (Kuba et al., 2022). To handle this problem, HAPPO (Kuba et al., 2022) proposes the multi-agent advantage decomposition lemma and sequential policy update scheme while introducing a multi-agent policy iteration procedure that enjoys the monotonic improvement guarantee. Inspired by the success of sequential decision-making strategy and the connection between sequential MARL and sequence models in nature language process (NLP), MAT (Wen et al., 2022) leverages a transformer to model the decision-making process. While HAPPO and MAT both utilize the PG algorithm, ACE (Li et al., 2022) focuses on value-based methods and proposes bi-directional action dependency to model the multi-agent decision-making process into a single-agent decision-making process by choosing actions for each agent one by one. With global information and action dependency, these methods successfully solve the non-stationarity problem and offer a monotonic improvement guarantee. In most benchmarks including SMAC, GRF, and Multi-Agent Mujoco, these methods significantly outperform strong CTDE baselines. This emphasizes the importance of global information and action-dependency for agents to cooperate with each other. However, in most scenarios, agents only take local observation as input and act synchronously in the execution phase, which means that the appliance of these sequential methods is limited compared to CTDE methods.

Also aiming to leverage more information in the decision-making phase, approaches based on agent modeling try to make agents predict other agents' information, including their observations, actions, and trajectories. He et al. (2016a) proposes a model that learns the behavior of opponents and automatically discovers different strategies of opponents. Grover et al. (2018) presents a domain-agnostic framework for agent modeling in multi-agent systems that requires only a few interaction episodes with other agents. In order to predict other agents' hidden goals, SOM (Raileanu et al., 2018) makes agents learn to infer other agents' hidden states from their behavior and choose actions based on the estimation. LIAM (Papoudakis et al., 2021) chooses to model other agents' trajectories from the trajectories of the controlled agent with a designed encoder-decoder structure. Although these methods also try to use local observations to predict more useful information, they focus mainly on modeling information about other agents, whose effect on the returns and goals of the controlled agent is indirect. In contrast, our prediction objective is the global representations of the centralized critic. These representations that will be used to estimate agents' state-action values are strongly related to the controlled agents' goals and rewards. Thus it is more convenient for agents to utilize.

In order to make agents predict the global representations from local observations, we need to add more layers to the agent network. However, as shown in D2RL (Sinha et al., 2020), increasing the number of layers in the network in reinforcement learning may lead to a decrease in sample efficiency and unstable performance for agents during training. D2RL introduces dense connections into RL algorithms by concatenating observation or state to each hidden layer of the network. We utilize both residual connections (He et al., 2016b) and dense connections in actor and critic network's layers to ensure a high sample efficiency and stable performance.

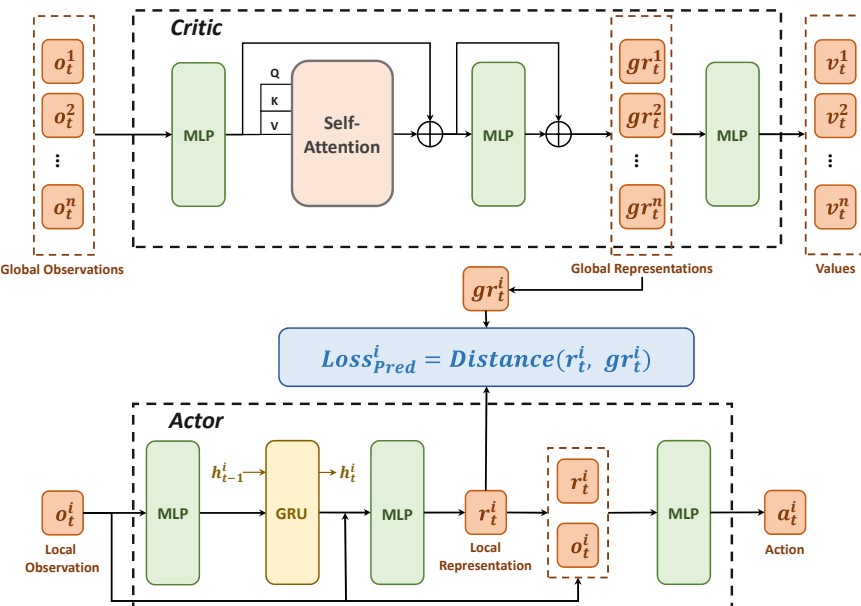

Figure 1: The actor-critic architecture of APC in CTDE framework. In the training phase, the centralized critic takes global observations from all the agents and utilizes the self-attention mechanism to generate global representations. On the one hand, the global representations are used to estimate the state-action values. On the other hand, these representations also serve as the ground truth for the actor to predict. The actor takes local observation and predicts the critic's global representations. The predicted representation is then concatenated with local observation and used to generate the next action. In the execution phase, only the actor works and chooses actions for agents with local observations as input.

# 3 METHODS

## 3.1 PRELIMINARIES

A decentralized partially observable Markov decision process (Dec-POMDP) (Oliehoek & Amato, 2016) could be defined as a tuple $G = <S, U, O, P, R, n, \gamma>$. $s \in S$ is the global state including the information of the environment and all the agents. $u \in U$ describes a joint action of $n$ agents. $o^i = O(s, i)$ means the local observation of agent $i$ when the global state is $s$. Given the joint action $u$, the probability of transition from state $s$ to $s'$ is described as $P(s'|s, u) : S \times U \times S \to [0, 1]$. $R(s, u) : S \times U \to \mathbb{R}$ denotes the reward function for agents. $n$ denotes the number of agents and $\gamma$ denotes the discount factor.

## 3.2 PREDICT GLOBAL REPRESENTATIONS FROM LOCAL OBSERVATIONS

In this section, we describe the method and the architecture (see Figure 1) of the proposed APC algorithm. Based on the actor-critic structure in the CTDE framework, APC consists of a centralized critic and a decentralized actor. At time step $t$, the critic takes global information as input and generates global representations $gr_t^1, gr_t^2...gr_t^n$ and state-action values $v_t^1, v_t^2...v_t^n$. The actor utilizes local observations $o_t^i$ to generate the predicted representation $r_t^i$ and then uses it to help generate the next action $a_t^i$. Both dense connections and residual connections are used to ensure that the information can pass through the network effectively and prevent the network from the gradient vanishing problem.

**Critic** Inspired by MAT, we leverage the self-attention mechanism (Vaswani et al., 2017) to encode the input into high-level global representations $gr_t^1, gr_t^2...gr_t^n$. Instead of using global states $s$ as input, we utilize the global observations $o_t^1, o_t^2...o_t^n$ that consist of a sequence of all the agents' observations as input. With the self-attention mechanism, the centralized critic can extract both the

global information and the interrelationships between the agents through the global observations while simply using the global state as input will make the generated global representations lack such relationships. Also, the representations generated from global observations are easier for agents to predict since the global states may contain unobservable information. Then, the global representations $gr_t^1, gr_t^2...gr_t^n$ that contain the global information and the interrelationships between agents are passed into the multi-layer perception (MLP) to estimate the state-action values.

Since the main purpose of the critic is to approximate the state-action value function for agents, the optimization objective of the critic is set as the clipped Bellman error:

$$Loss_{critic}(\phi) = \frac{1}{nT} \sum_{t=1}^{T} \sum_{i=1}^{n} max[(V_\phi(s_t^i) - \hat{R}_t)^2, CLIP^2], \tag{1}$$

where $\phi$ denotes the parameters of the critic network. $t$ refers to the current environment time step. $CLIP$ denotes the Bellman error after being clipped:

$$CLIP = clip(V_\phi(s_t^i), V_{\phi_{old}}(s_t^i) - \varepsilon, V_{\phi_{old}}(s_t^i) + \varepsilon) - \hat{R}_t, \tag{2}$$

where $V_{\phi_{old}}(s_t^i)$ denotes the old value estimation from data batch. $\varepsilon$ denotes the clip parameter and $\hat{R}_t$ denotes the discounted reward-to-go:

$$\hat{R}_t = R_t + \gamma V_\phi(s_{t+1}^i). \tag{3}$$

**Actor**  We utilize parameter sharing in APC, which means all the agents share the same critic and actor networks. This will increase training efficiency and save compute costs (Christianos et al., 2021). To better leverage the historical observations to predict global representations, the actor uses GRU (Chung et al., 2014) to encode local observation $o_t^i$ and hidden state $h_{t-1}^i$ of last time step into new hidden state $h_t^i$ and representation $r_t^i$ which serves as the prediction of global representation. The predicted representation is then concatenated with the local observation and passed into MLP to output the probability distribution of the current agent, which is denoted as $\pi_\theta^i(a_t^i|o_t^i)$. Here $\theta$ denotes the parameters of the actor network.

In order to train the actor to predict global representations and make decisions for agents, the objective for the actor to minimize is set as follows:

$$Loss_{actor}(\theta) = Loss_{policy}(\theta) + \sigma Loss_{entropy}(\theta) + \beta Loss_{prediction}(\theta). \tag{4}$$

The loss for the actor consists of three parts: policy loss, entropy loss, and prediction loss. $\sigma$ and $\beta$ denote the coefficient hyper-parameters of entropy and prediction loss relatively. The policy loss aims to optimize the clipping PPO objective:

$$Loss_{policy}(\theta) = \frac{1}{nT} \sum_{t=1}^{T} \sum_{i=1}^{n} min(r_t^i(\theta)\hat{A}_t^i, clip(\bar{r}_t^i(\theta), 1 \pm \varepsilon)\hat{A}_t^i), \tag{5}$$

where

$$\bar{r}_t^i(\theta) = \frac{\pi_\theta^i(a_t^i|o_t^i)}{\pi_{\theta_{old}}^i(a_t^i|o_t^i)}, \tag{6}$$

and $\hat{A}_t^i$ denotes the advantage estimated by the GAE (Schulman et al., 2015) method.

The entropy loss is used to ensure the sufficient exploration (Schulman et al., 2017):

$$Loss_{entropy}(\theta) = \frac{1}{nT} \sum_{t=1}^{T} \sum_{i=1}^{n} E[\pi_\theta(o_t^i)], \tag{7}$$

Table 1: Performance of APC and other baselines on the SMAC benchmark. We collect the average win rates and standard deviations of each method in the last 5 evaluation rounds over 10 random seeds.

| Task | Difficulty | APC | MAT-Dec | MAPPO | QMIX | Steps |
|---|---|---|---|---|---|---|
| 8m | Easy | 99.2(1.0) | 97.5(1.0) | **99.7**(0.3) | 98.2(1.0) | 1e6 |
| MMM | Easy | 99.5(1.5) | **99.7**(0.9) | 99.5(0.6) | 99.4(0.2) | 2e6 |
| 8m vs 9m | Hard | **98.5**(1.6) | 95.1(1.9) | 93.9(0.8) | 95.2(3.0) | 5e6 |
| 2c vs 64zg | Hard | **97.5**(2.0) | 94.8(4.2) | 93.5(2.6) | 96.6(0.5) | 5e6 |
| MMM2 | Hard+ | **94.4**(3.3) | 91.4(3.6) | 91.1(5.2) | 86.9(0.3) | 1e7 |
| 6h vs 8z | Hard+ | **81.8**(7.1) | 72.4(26.1) | 74.3(18.7) | 70.7(5.9) | 1e7 |
| 3s5z vs 3s6z | Hard+ | **89.6**(5.8) | 84.5(9.9) | 69.2(9.8) | 74.0(1.4) | 2e7 |

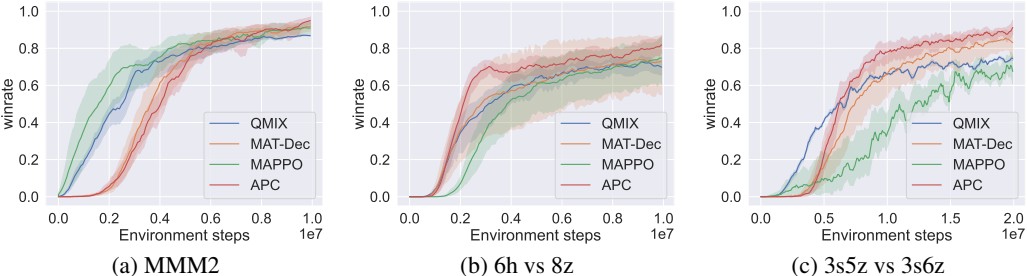

(a) MMM2      (b) 6h vs 8z      (c) 3s5z vs 3s6z

Figure 2: Performance comparison of APC and other baselines on the hard+ SMAC tasks. We conduct experiments on each map over 10 random seeds. The results show that APC significantly outperforms other baselines on hard+ tasks where local information is not enough to reach perfect cooperation.

where $E$ denotes the entropy bonus.

**Prediction Loss** Prediction loss is used to train the actor to predict the global representations generated by the centralized critic. So we set the prediction loss as follows to measure the distance between the predicted representation and the global representation of the critic:

$$Loss_{prediction}(\theta) = \frac{1}{nT} \sum_{t=1}^{T} \sum_{i=1}^{n} Distance(r_t^i, gr_t^i), \quad (8)$$

where $Distance$ denotes the distance between the predicted representation generated by the actor and the global representation generated by the critic. We simply use the *Euclidean Distance* in this work. During the training, the actor learns to minimize the prediction loss to predict the global information. The introduction of prediction loss also helps the actor learn directly from the centralized critic's representation rather than only from the estimated state-action values. This enables the algorithm to make better use of the global information in the training phase.

## 4  EXPERIMENTS

To demonstrate the advantage of APC compared with other CTDE methods, we test APC and several strong baselines in CTDE framework on various benchmarks, including StarCraft Multi-Agent Challenge (SMAC) (Samvelyan et al., 2019), Google Research Football (GRF) (Kurach et al., 2020) and Multi-Agent Mujoco (Peng et al., 2021). These benchmarks are widely used in MARL as cooperation tasks. To ensure the credibility of the comparison, we choose both the state-of-the-art (SOTA) PG and VD methods as baselines, including MAT-Dec (the CTDE version of MAT), MAPPO, and fine-tuned QMIX. The parameters of these algorithms are set as suggested in their original papers

Table 2: Statistical significance of the results on SMAC hard+ maps. We compare the results of APC and other baselines using the t-test (independent two-sample t-test). Here we collect the P-values that measure the significance of the advantages of APC against other baselines.

| Task | APC | MAT-Dec | MAPPO | QMIX |
|---|---|---|---|---|
| MMM2 | - | 1.89e-05 | 2.11e-04 | 1.76e-22 |
| 6h vs 8z | - | 2.80e-02 | 2.08e-02 | 2.54e-12 |
| 3s5z vs 3s6z | - | 1.56e-03 | 4.82e-22 | 8.93e-27 |

to achieve their best performance. Fine-tuned QMIX is only tested on SMAC since its parameters have only been fine-tuned on SMAC. For APC, we empirically set the coefficient hyper-parameters $\sigma = 0.01$ and $\beta = 0.01$. For the details of hyper-parameter setting and results of all the experiments please see Appendix B and C.

### 4.1 RANDOM SEED SETTINGS AND STATISTICAL SIGNIFICANCE TESTS

For each task, we conduct experiments with 10 different random seeds. In the tables of results, we report the average win rates/rewards/scores on the last 5 evaluation rounds over all the 10 random seeds to prevent the effect of randomness on the results. Meanwhile, due to the large variance that some algorithms show in some experiments with different random seeds, we conduct statistical significance tests on the results of APC and other algorithms to make our conclusions more convincing. We utilize the t-test (independent two-sample t-test) to show the significance of APC's advantages against other baselines. The P-values show the significance of the difference between results. The lower the P-value is, the more significant the gap between the results of APC and other baselines is (For example if the P-value is lower than 0.05, we can say APC is significantly better than other methods with a 95% CI). For the results of all the statistical significance experiments please see Appendix D.

### 4.2 PERFORMANCE ON SMAC BENCHMARK

For the SMAC benchmark, we test APC and other baselines on several maps and the maps' difficulty ranges from easy to hard+. Table 1 collects the average win rates of these methods against the games' built-in AI. The results show that APC achieves the highest average win rates on most maps compared with other baselines. In simple scenarios, the advantage of APC is not particularly obvious, as the cooperation here is easy to accomplish even without global information for agents. But in the hard+ scenarios, since the P-values (see in Table 2) are lower than 0.05, APC significantly outperforms other methods, such as map *3s5z vs 3s6z* shown in Figure 2c, which indicates the importance of the information contained in the global representations to achieve cooperation.

On the hard and hard+ tasks, we find that APC also performs very well on the stability over the ten random seeds. Despite utilizing self-attention mechanism like MAT-Dec and thus having a larger number of parameters and network depth, APC has much less variance in the results than MAT-Dec and even MAPPO, second only to QMIX, which has the smallest number of parameters (see in Figure 1, especially in map *6h vs 8z*, where the variance of MAT-Dec and MAPPO are 26.1 and 18.7, while the variance of APC is only 7.1). This further validates our claim that in APC the actor network not only learns from the state action values given by the critic, but also the information in global representations that are used to generate these values. Since the training of the critic is much easier than the actor, learning more information from the critic about agents' goals and rewards also helps the stability of the training of the actor under different random seeds.

### 4.3 PERFORMANCE ON GRF AND MULTI-AGENT MUJOCO BENCHMARKS

To make the conclusions more convincing, we also conducted experiments on the GRF and Multi-Agent Mujoco benchmarks. In GRF, the local observations of agents contain a lot of information, so the key problem for agents is how to learn the interrelationships between each other and cooperate to win high scores. Although both APC and MAT-Dec model such relationships using the self-attention mechanism in the critic, the actor in MAT-Dec only learns from the critic's estimation of

the state value. But in APC, the actor can also learn the representations extracted by the critic, which contains the high-level features that are used to estimate the action values. The results on GRF (see in Table 3) are consistent with our analysis and APC obviously gains the highest average scores in all three scenarios. In Multi-Agent Mujoco, agents only observe their own and neighbors' states, which means that agents could benefit a lot from the information contained in predicted representations. The results of the experiment also confirm the above instinct. As shown in Table 4 and Figure 3, APC significantly outperforms these strong CTDE baselines in average rewards.

Table 3: Performance of APC and other baselines on the Google Research Football benchmark. We collect the average scores and standard deviations of each method in the last 5 evaluation rounds over 10 random seeds.

| Task | APC | MAT-Dec | MAPPO | Steps |
|---|---|---|---|---|
| pass and shoot with keeper | **0.967**(0.028) | 0.951(0.032) | 0.923(0.051) | 5e6 |
| 3 vs 1 with keeper | **0.947**(0.038) | 0.930(0.042) | 0.736(0.104) | 5e6 |
| counterattack easy | **0.938**(0.044) | 0.902(0.049) | 0.733(0.101) | 1e7 |

Table 4: Performance of APC and other baselines on the Multi-Agent Mujoco benchmark. We collect the average rewards and standard deviations of each method in the last 5 evaluation rounds over 10 random seeds.

| Task | APC | MAT-Dec | MAPPO | Steps |
|---|---|---|---|---|
| HalfCheetach 6x1 | **6923**(894) | 6108(1150) | 1701(607) | 1e7 |
| HalfCheetach 2x3 | **6985**(608) | 6491(792) | 3040(580) | 1e7 |

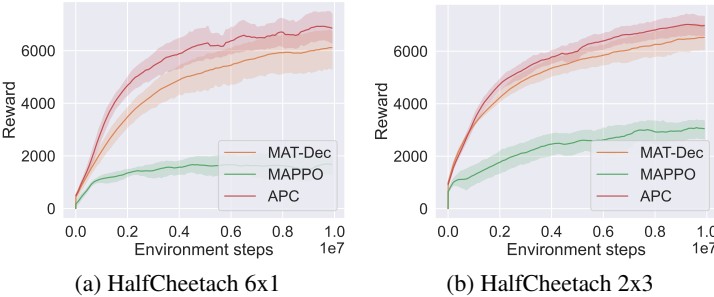

(a) HalfCheetach 6x1    (b) HalfCheetach 2x3

Figure 3: Performance comparison of APC and other baselines on the Multi-Agent Mujoco tasks.

### 4.4 ABLATION STUDIES

As the actor of APC utilizes additional layers to predict global representations, to demonstrate that the improvement of performance comes from the predicted representations rather than a deeper network, we set the coefficient $\beta$ of prediction loss to 0 and compare its performance with the original APC. The results (see in Figure 4) on SMAC and Multi-Agent Mujoco show that the performance of APC becomes extremely worse without prediction loss. This poor performance is most likely due to the fact that with the number of layers increasing, it becomes difficult for the network to deliver the forward observation information and the backward gradient information (Sinha et al., 2020), even if we already use dense connections and residual connections to avoid such problem. The results obviously demonstrate the effectiveness of predicting global representations.

Moreover, to illustrate the importance of utilizing historical information to bridge the gap between local and global information in predicting, we replace GRU with MLP in the prediction part of the actor as a comparison. We collect the performance of these two versions of APC in Figure 5, where the APC with GRU obviously outperforms the APC with MLP. This indicates that without histor-

ical information, it's difficult to bridge the gap between global information and local information. Therefore, the use of historical information in APC has proved to be necessary and effective.

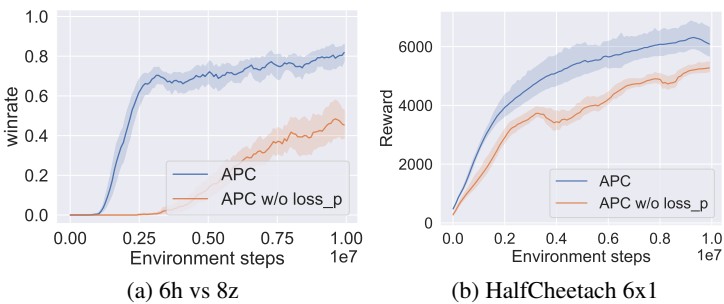

(a) 6h vs 8z       (b) HalfCheetach 6x1

Figure 4: Performance comparison of APC and APC without prediction loss on the 6h vs 8z task of SMAC benchmark and HalfCheetach 6x1 task of Multi-Agent Mujoco benchmark.

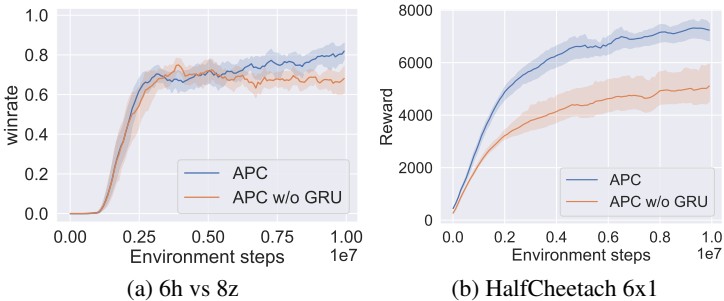

(a) 6h vs 8z       (b) HalfCheetach 6x1

Figure 5: Performance comparison of APC and APC without GRU (replaced by MLP) on the 6h vs 8z task of SMAC benchmark and HalfCheetach 6x1 task of Multi-Agent Mujoco benchmark.

## 5    CONCLUSION

In this paper, we propose the actor-predicts-critic (APC) based on the actor-critic architecture in the CTDE framework. In APC, the actor learns to predict the global representations with local observation in the training phase and utilizes the information in the predicted representations to achieve better cooperation in the execution phase. The experiments on various benchmarks show that APC significantly outperforms the strong baselines in the CTDE framework including MAT-Dec, MAPPO, and fine-tuned QMIX. Furthermore, the ablation studies demonstrate the validity of the proposed APC's architecture and the effectiveness of utilizing the predicted global information to cooperate. However, there are still some limitations to this work that are left for further study. Since action dependency is unavailable under the CTDE framework, there is still a small gap between the performance of APC and the sequential decision-making methods with centralized controllers. With the predicted global representations, APC could only mitigate the non-stationarity problem rather than completely avoid it. Moreover, In addition to the information in the global representations, there is a lot of kinds of information waiting to be discovered and utilized. In the future, we plan to enable agents to implicitly benefit from action dependency by introducing priority information for each agent. Also, we will theoretically analyze the impact of different kinds of information on multi-agent cooperation and find a way to better utilize global information during training. We hope to completely prevent agents in the CTDE framework from suffering the non-stationarity problem and make them learn better cooperation strategies in future studies.

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

APPENDIX

# A DISCUSSION ABOUT PREDICTING ACTIONS OF OTHERS IN CTDE FRAMEWORK

In this section, we will illustrate that predicting other agents' actions and choosing actions based on the prediction in the CTDE framework makes no sense. For example, suppose there is a simple game with only two agents ($A_1$ and $A_2$) participating. At each time step, agents choose a number from $\{-1, 1\}$. If agents choose the same number, they win. If agents choose a different number, they lose. At time step $t-1$, suppose $A_1$ chooses $-1$, $A_2$ chooses 1, and according to the rules they lose this round. Then, at time step $t$, if they only observe the choices made in the last turn and can not predict each others' new decisions, $A_1$ will choose 1 and $A_2$ will choose $-1$, which leads to another failure. However, if they can accurately predict each others' decisions, based on the prediction of each other's decision, $A_1$ will choose $-1$ and $A_2$ will choose 1, which still leads to another failure. The reason is that the agent's prediction of others fails when it makes a decision and other agents change their decisions based on the prediction of this agent. This process is called the "k-level reasoning" in game theory. So as long as the agents make decisions synchronously, it is meaningless and self-contradictory to predict the actions of others and act according to the prediction.

# B HYPER-PARAMETERS USED FOR DIFFERENT METHODS

## B.1 PPO METHODS

Table 5: Common hyper-parameters used for APC, MAT-Dec and MAPPO in the SMAC domain

| hyper-parameters | value | hyper-parameters | value |
|---|---|---|---|
| critic lr | 5e-4 | optimizer | Adam |
| actor lr | 5e-4 | optim eps | 1e-5 |
| training threads | 16 | max grad norm | 10 |
| rollout threads | 32 | use huber loss | True |
| hidden layer dim | 64 | batch size | 3200 |

Table 6: Hyper-parameters used for APC and MAT-Dec in the SMAC domain

| Maps | ppo epochs | ppo clip | $\gamma$ | Steps |
|---|---|---|---|---|
| 8m | 15 | 0.2 | 0.99 | 1e6 |
| MMM | 15 | 0.2 | 0.99 | 2e6 |
| 8m vs 9m | 10 | 0.05 | 0.99 | 5e6 |
| 2c vs 64zg | 10 | 0.05 | 0.99 | 5e6 |
| MMM2 | 5 | 0.05 | 0.99 | 1e7 |
| 6h vs 8z | 15 | 0.05 | 0.99 | 1e7 |
| 3s5z vs 3s6z | 5 | 0.05 | 0.99 | 2e7 |

Table 7: Hyper-parameters used for MAPPO in the SMAC domain

| Maps | ppo epochs | ppo clip | $\gamma$ | Steps |
|---|---|---|---|---|
| 8m | 15 | 0.2 | 0.99 | 1e6 |
| MMM | 15 | 0.2 | 0.99 | 2e6 |
| 8m vs 9m | 15 | 0.05 | 0.99 | 5e6 |
| 2c vs 64zg | 5 | 0.2 | 0.99 | 5e6 |
| MMM2 | 5 | 0.2 | 0.99 | 1e7 |
| 6h vs 8z | 5 | 0.2 | 0.99 | 1e7 |
| 3s5z vs 3s6z | 5 | 0.2 | 0.99 | 2e7 |

Table 8: Common hyper-parameters used in the Google Research Football domain

| hyper-parameters | value | hyper-parameters | value |
|---|---|---|---|
| critic lr | 5e-4 | $\gamma$ | 0.99 |
| actor lr | 5e-4 | optimizer | Adam |
| steps | 1e-7 | optim eps | 1e-5 |
| training threads | 16 | max grad norm | 10 |
| rollout threads | 40 | use gae | True |
| episode length | 100 | use huber loss | True |
| hidden layer dim | 64 | batch size | 3200 |

Table 9: Different hyper-parameters used in the Google Research Football domain

| Maps | APC | MAT-Dec | MAPPO |
|---|---|---|---|
| ppo epochs | 10 | 10 | 5 |
| ppo clip | 0.05 | 0.05 | 0.2 |
| num hidden layer | 1 | / | 2 |

Table 10: Common hyper-parameters used in the Multi-Agent Mujoco domain

| hyper-parameters | value | hyper-parameters | value |
|---|---|---|---|
| $\gamma$ | 0.99 | optimizer | Adam |
| steps | 1e-7 | optim eps | 1e-5 |
| training threads | 16 | max grad norm | 10 |
| rollout threads | 40 | use gae | True |
| episode length | 100 | use huber loss | True |
| hidden layer dim | 64 | batch size | 3200 |

Table 11: Different hyper-parameters used in the Multi-Agent Mujoco domain

| Maps | APC | MAT-Dec | MAPPO |
|---|---|---|---|
| critic lr | 5e-5 | 5e-5 | 5e-3 |
| actor lr | 5e-5 | 5e-5 | 5e-6 |
| ppo epochs | 10 | 10 | 5 |
| ppo clip | 0.05 | 0.05 | 0.2 |
| num hidden layer | 1 | / | 2 |

## B.2 VALUE-BASED METHODS (FINE-TUNED QMIX)

Please refer to `https://github.com/hijkzzz/pymarl2`

# C RESULTS OF EXPERIMENTS

In this section, we will clearly show the results of all the experiments on SMAC, GRF and Multi-Agent Mujoco benchmarks.

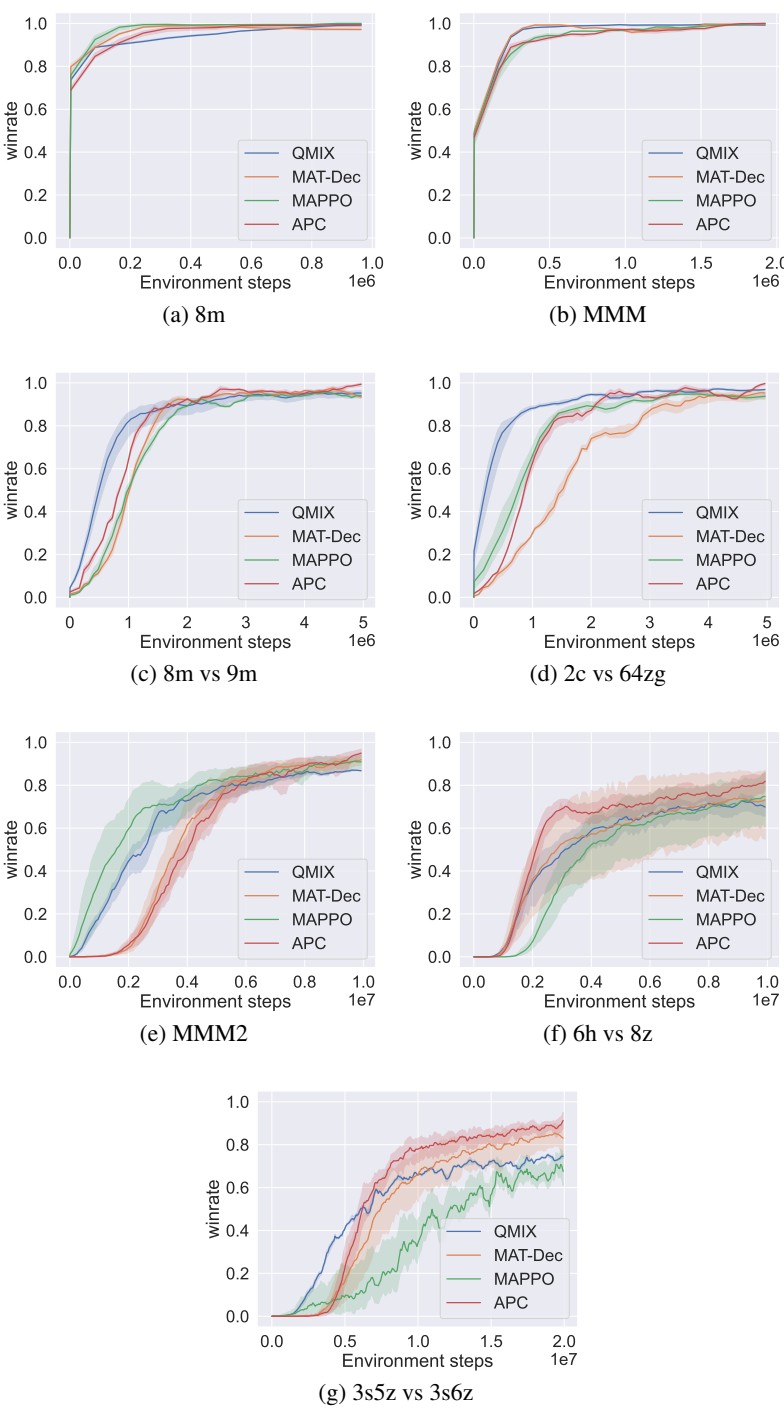

Figure 6: Performance comparison of APC and other baselines on the SMAC benchmark. We collect the average win rates and standard deviations of each method over 10 random seeds.

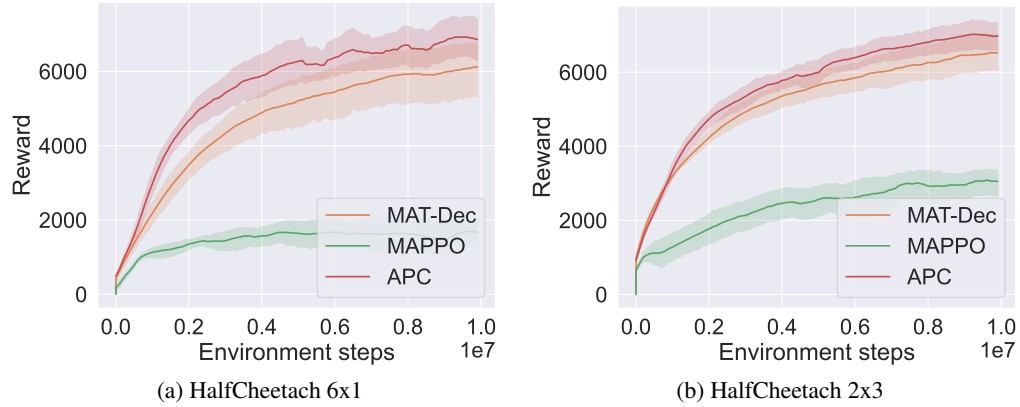

(a) HalfCheetach 6x1

(b) HalfCheetach 2x3

Figure 7: Performance comparison of APC and other baselines on the Multi-Agent Mujoco benchmark. We collect the average rewards and standard deviations of each method over 10 random seeds.

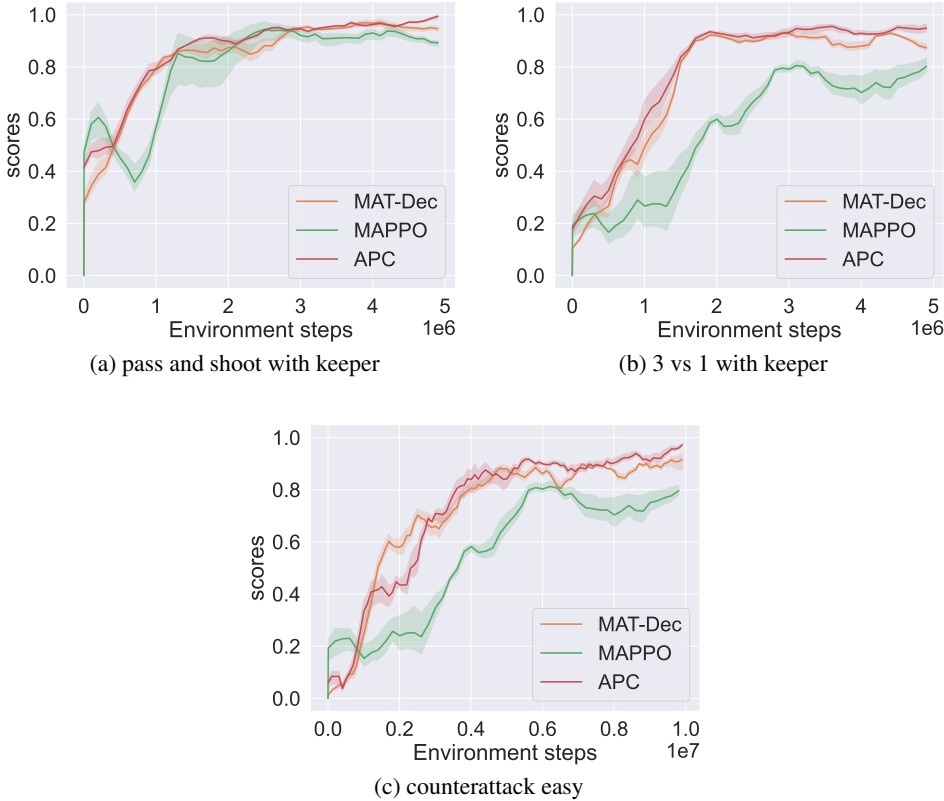

(a) pass and shoot with keeper

(b) 3 vs 1 with keeper

(c) counterattack easy

Figure 8: Performance comparison of APC and other baselines on the Google Research Football benchmark. We collect the average scores and standard deviations of each method over 10 random seeds.

## C.1 ABLATION STUDIES

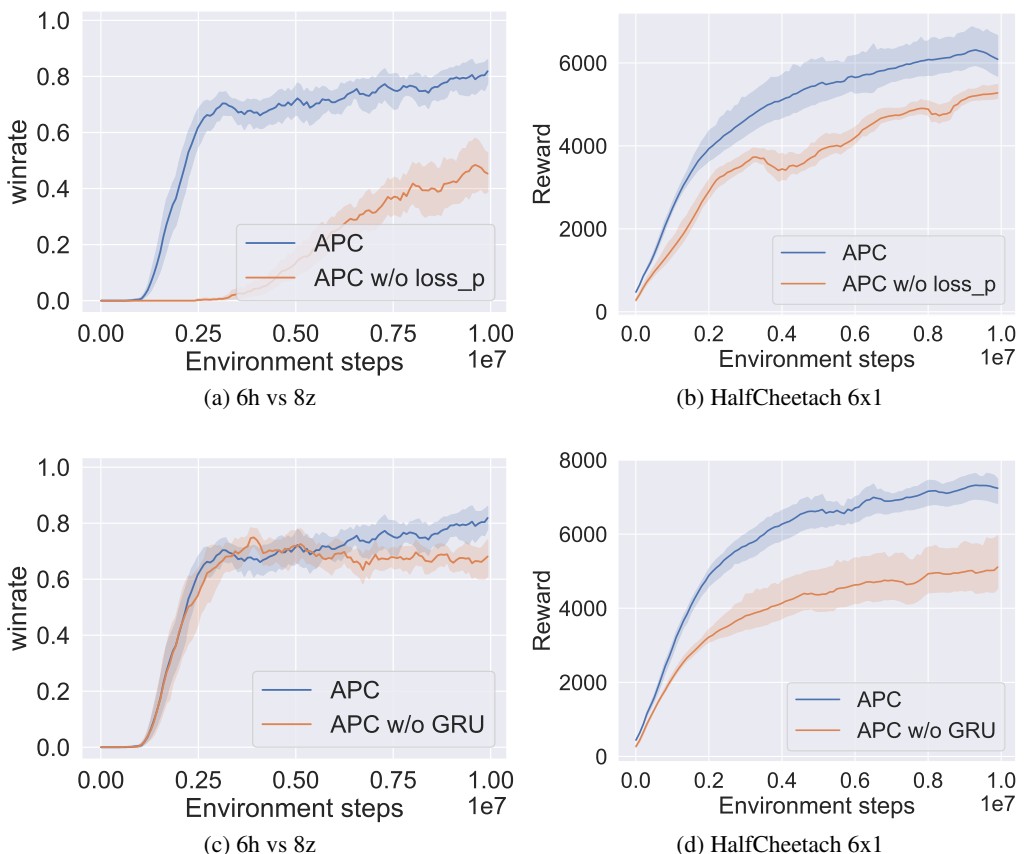

Figure 9: Performance comparison of APC, APC without prediction loss and APC without GRU (replaced by MLP). We conduct experiments of each method on each map over 10 random seeds.

## D    STATISTICAL SIGNIFICANCE RESULTS

Table 12: Statistical significance of the results on SMAC hard and hard+ maps. We compare the results of APC and other baselines using the t-test (independent two-sample t-test). Here we collect the P-values that measure the significance of the advantages of APC against other baselines.

| Task | APC | MAT-Dec | MAPPO | QMIX |
|---|---|---|---|---|
| 8m vs 9m | - | 9.92e-17 | 3.55e-31 | 2.53e-10 |
| 2c vs 64zg | - | 5.96e-05 | 7.52e-21 | 2.99e-03 |
| MMM2 | - | 1.89e-05 | 2.11e-04 | 1.76e-22 |
| 6h vs 8z | - | 2.80e-02 | 2.08e-02 | 2.54e-12 |
| 3s5z vs 3s6z | - | 1.56e-03 | 4.82e-22 | 8.93e-27 |

Table 13: Statistical significance of the results on GRF benchmark. We compare the results of APC and other baselines using the t-test (independent two-sample t-test). Here we collect the P-values that measure the significance of the advantages of APC against other baselines.

| Task | APC | MAT-Dec | MAPPO |
|---|---|---|---|
| P&S with keeper | - | 1.24e-02 | 8.32e-07 |
| 3 vs 1 with keeper | - | 2.69e-02 | 1.83e-21 |
| counterattack easy | - | 1.31e-04 | 1.41e-21 |

Table 14: Statistical significance of the results on Multi-agent Mujoco benchmark. We compare the results of APC and other baselines using the t-test (independent two-sample t-test). Here we collect the P-values that measure the significance of the advantages of APC against other baselines.

| Task | APC | MAT-Dec | MAPPO |
|---|---|---|---|
| HalfCheetach 6x1 | - | 9.41e-05 | 1.20e-55 |
| HalfCheetach 2x3 | - | 5.03e-04 | 6.00e-59 |

