# OpenReview forum: "APC: Predict Global Representation From Local Observation In Multi-Agent Reinforcement Learning"
_ICLR.cc/2024/Conference — Submitted to ICLR 2024_

### Official Review · Reviewer_iCg2 · 2023-10-17

**Soundness:** 2 fair
**Presentation:** 2 fair
**Contribution:** 2 fair
**Rating:** 5
**Confidence:** 4

**Summary:**

This paper utilizes actors to predict global representations from local observations. During the training process, the estimated state values are received and the loss error between the global representation of the reviewer and the predicted representation is established.

**Strengths:**

Utilizing the global information representation in critic as the prediction target allows local observation agents to have predictive judgments on the global state. This idea is novel and has a certain degree of innovation.

**Weaknesses:**

There is no proof of the theoretical feasibility of the proposed algorithm.  The experiment is relatively simple.

**Questions:**

While updating the critic network, the gradient generated by prediction loss in the algorithm will update the state representation. Is this training stable?

How to ensure convergence? What is the loss error compared to learning during training?

---

> ### Author Response · Authors · 2023-11-18
>
> We wish to thank reviewer iCg2 for carefully reviewing our paper and for the constructive comments! Please see below our responses. We use ***Q***, ***W***, ***A*** to denote question, weakness, and answer.
>
> >***Q1***
>
> ***A1***: We block the gradient update of the global representations provided by the critic when calculating the prediction error to ensure that the critic’s update only considers the error of estimating the state-action value.
>
> >***Q2***
>
> ***A2***: For the actor network, we only added a prediction module to learn the global representations and provide additional information for the subsequent decision-making process. So it will not affect the convergence of the network.

---

### Official Review · Reviewer_2e7K · 2023-10-23

**Soundness:** 1 poor
**Presentation:** 2 fair
**Contribution:** 1 poor
**Rating:** 3
**Confidence:** 5

**Summary:**

This paper proposes addressing partial observability by utilizing partial observations to predict the global representation for the critic, and using the learned representation as an additional input to assist in algorithm learning for the actor. The resulting algorithm ultimately outperforms some baseline algorithms on GRF, Multi-Agent Mujoco, and SMAC tasks.

**Strengths:**

In MARL, addressing the issue of partial observability is a pivotal research concern. The approach adopted in this paper, which is based on predicting global states from local observations, holds meaningful potential. However, the proposed method introduces several questions that require further elucidation and discussion.

**Weaknesses:**

- The primary concern in this paper revolves around the significance of predicting the representation in the critic. The representations in the critic undergo drastic changes, exhibit high levels of instability, and lack interpretability. The author provides intuitive reasons for its effectiveness but lacks theoretical support and experimental analysis. Furthermore, due to the complexity of the algorithm architecture, including the use of GRU, attention mechanisms, and complex AC network structure, analyzing why this method is effective becomes more challenging.
- There are many approaches to addressing the issue of partial observability, such as predicting the global state from local observations. However, this paper lacks a thorough investigation into such methods, and there is no discussion of related work in this regard.
- The experimental evaluation in this paper is relatively weak. Firstly, the algorithm does not appear to converge. For instance,  fine-tuned QMIX can already achieve a 100% win rate on almost all SMAC tasks by itself. Can APC achieve better performance after sufficient training? Additionally, the experiments on MA MuJoCo only cover two tasks, and the performance of on-policy baselines in such tasks is not effective. The author should consider comparing with off-policy MARL algorithms. In addition, this paper is not compared to related works addressing the same issue. Certainly, this depends on a thorough investigation of related works, which is also a crucial aspect lacking in this paper.
- Finally, in the appendix, the authors argue that predicting other agents' behaviors in the CTDE framework is not meaningful. I disagree with this viewpoint. Since the learning process in MARL is dynamic and the optimization direction is aligned, guided learning through policy gradients leads the agent's behavior to converge to suboptimal states. At this point, being able to make accurate estimates of other agents' behavior is likely to have a positive impact on the learning process.

**Questions:**

- Why do we need to the loss p? Directly sharing the critic's input with the actor might be a more straightforward approach.
- Intuitively, it seems that the proposed predictive loss in this paper should not rely on such complex architectural designs. Only through a comparison under identical architectures can it be demonstrated whether the proposed loss in this paper is truly effective. Would there be a performance gain when comparing under the same structure?
- Could the authors provide a performance comparison on different tasks using the same network architecture as the baseline?

---

> ### Author Response · Authors · 2023-11-18
>
> We wish to thank reviewer 2e7K for carefully reviewing our paper and for the constructive comments! Please see below our responses. We use ***Q***, ***W***, ***A*** to denote question, weakness, and answer.
>
> >***Q1***
>
> ***A1***: Sharing the global information provided by the critic directly with the actor would break the CTDE framework. This will make the algorithm centralized and require global information as input during the execution phase, which is not allowed in many application scenarios. (This point has been mentioned in the Introduction section.)
>
> >***Q2*** & ***W1***
>
> ***A2***: In fact, our network structure is not as complicated as you described. The structure of the critic network is also used by MAT (MAT-Dec as well). In the actor network, we only added a GRU module for learning global representation compared to MAT-Dec.
>
> >***Q3***
>
> ***A3***: As ***A2*** has mentioned, our network is essentially based on MAT-Dec with the addition of a representation prediction module based on GRU. And the experimental results do show significant improvement compared to MAT-Dec which proves the effectiveness of APC. However, we still appreciate the reviewer’s suggestion and will further try to add the representation prediction module to MAPPO and observe the experimental results.
>
> >***W2***
>
> ***A4***: We have already introduced some relevant methods about predicting additional information in the Introduction section. However, most of the algorithms we have researched have no open-source code or have not been tested on the SMAC environment. Reproducing these algorithms and finding the best parameters for these algorithms in the SMAC environment requires a lot of time and effort. Therefore, we did not include these methods in the comparison. We will investigate more methods and conduct experiments to compare them with APC in future work.
>
> >***W4***
>
> ***A5***: As HAPPO[1] has already pointed out, if the agents make synchronous decisions and there is no sequential dependency between their actions, it might not lead to the optimal result.
>
> [1] Jakub Grudzien Kuba, Ruiqing Chen, Muning Wen, Ying Wen, Fanglei Sun, Jun Wang, and Yaodong Yang. Trust region policy optimization in multi-agent reinforcement learning. In International Conference on Learning Representations, 2022.

---

### Official Review · Reviewer_XhHc · 2023-10-25

**Soundness:** 2 fair
**Presentation:** 3 good
**Contribution:** 2 fair
**Rating:** 5
**Confidence:** 3

**Summary:**

The paper proposes the actor-predicts-critic (APC) algorithm to make the actor predict the global presentations of the critic from local observations. The authors claim that predicting the global presentations of the critic helps agents choose optimal action by mitigating the non-stationary problem. APC utilizes self-attention to generate the global presentations for each agent’s actor and training actor to minimize the prediction loss. Experiments on SMAC, Google Research Football, and Multi-Agent Mujoco validate the effectiveness of APC.

**Strengths:**

1.	This paper is well-written and well-structured. The idea of predicting the critic’s global presentations is straightforward and easy to understand.
2.	APC shows improvements on three benchmarks over strong baselines including MAPPO and fine-tuned QMIX and statistical significance tests are conducted.

**Weaknesses:**

1.	Using auxiliary tasks to help the actor’s decision-making is not a new thing. Although the authors claim that predicting the critic’s global presentations is better than other forms, there is no theoretical or experimental evidence in the paper to support this claim.
2.	Besides adding the predicting loss, the authors also utilize the residual connections in the network design, the authors do not show the ablation study of this trick and whether it could boost the performance of the compared baselines.
3.	The performance improvements are slight, especially compared with MAT-Dec. In most cases, the performances of APC and MAT-Dec are very similar.

**Questions:**

1.	Could the authors provide the parameter numbers of APC and the compared baselines?
2.	The reported performance of fine-tuned QMIX in this paper is not consistent with the original paper. For example, on the MMM2 scenario in SMAC, the authors report the test win rate as 86.9% in this paper while 100% is reported in the fine-tuned QMIX’s original report. Could the authors clarify any difference between their setting and the fine-tuned QMIX’s original setting?
3.	Could the authors provide the ablation studies of the number of network layers and residual connections?
4.	Could the authors provide more explanations about the choice of Euclidean Distance as the distance metric?
5.	How does the performance of predicting the critic’s global presentations compare with other auxiliary tasks?

---

> ### Author Response · Authors · 2023-11-18
>
> We wish to thank reviewer XhHc for carefully reviewing our paper and for the constructive comments! Please see below our responses. We use ***Q***, ***W***, ***A*** to denote question, weakness, and answer.
>
> >***Q1***
>
> ***A1***: Here are the parameter numbers (on the 6h vs 8z map) of APC and the compared baselines:
>
> | APC | MAT-Dec | MAPPO | QMIX |
> | -------- | -------- | -------- | -------- |
> | 116513 | 60361 | 72553 | 73871 |
>
> Most of the additional parameters of APC come from the prediction module in the actor, which is mainly based on GRU.
>
> >***Q2***
>
> ***A2***: We believe that the difference in fine-tuned QMIX’s performance comes from two aspects: random seed settings and measurement methods. Since the fine-tuned QMIX does not provide detailed seed settings, we use our own generated random seeds, which may lead to differences between the results. Additionally, the evaluation metric used by fine-tuned QMIX is the best median win rate. In our paper, we use the average win rates of each method in the last 5 evaluation rounds over 10 random seeds.
>
> >***Q3***
>
> ***A3***: Due to time constraints, we only conducted ablation experiments about the number of layers on the 6h vs 8z map in SMAC, and the results are as follows:
>
> | APC with mlp1 | APC with mlp3 | APC with mlp5 |
> | -------- | -------- | -------- |
> | 66.7(3.3) | 81.8(7.1) | 72.3(3.6) |
>
> We changed the number of layers in the MLP of the actor (with 1, 3, and 5 layers). It can be seen that as the number of layers in the network increased from 1 to 3, the winning rate increased significantly. However, when the number of layers increased to 5, the network’s performance slightly decreased, which is consistent with the point mentioned in this paper that information transmission and gradient updates become more difficult as the number of layers in the network increases. We believe that the ablation experiments in the article are sufficient to attribute the performance of APC to global representation prediction rather than changes in network layers or tricks like residual connections.
>
> >***Q4***
>
> ***A4***: We tested both the Euclidean distance and the MSE loss function, and found no significant difference between these two. So we used the Euclidean distance as the loss function.
>
> >***Q5***
>
> ***A5***: Unfortunately, most of the algorithms we have researched have no open-source code or have not been tested on the SMAC environment. Reproducing these algorithms and finding the best parameters for these algorithms in the SMAC environment requires a lot of time and effort. Therefore, we did not include these methods in the comparison. We will investigate more methods and conduct experiments to compare them with APC in future work.

---

### Official Review · Reviewer_HFwe · 2023-10-28

**Soundness:** 3 good
**Presentation:** 3 good
**Contribution:** 2 fair
**Rating:** 5
**Confidence:** 3

**Summary:**

This paper proposed the actor-predicts-critic (APC) algorithm, in which the actor learns to predict the global representations of a centralized critic from local observation. Since these global representations are closely related to agents' goals and rewards, agents can achieve better cooperation on MARL tasks. To validate APC, this paper evaluated the algorithm on StarCraft2, Google Research Football, and Multi-Agent Mujoco benchmarks. The results showed that APC significantly outperformed the strong baselines in centralized training and decentralized execution (CTDE) framework, including MAT-Dec, MAPPO, and fine-tuned QMIX.

**Strengths:**

The paper provides the APC algorithm to learn to predict the global representations of a centralized critic from local observation, which is based on a simple but reasonable idea to consider the loss function to minimize the distance between local and global representation.
The experimental results based on well-known benchmarks showed that APC outperformed other baselines in terms of average rewards.

**Weaknesses:**

The methodological contribution other than predicting the global representations (i.e., Loss_{prediction}) was unclear to me. Although the idea of APC is simple and effective, comparison with CTDE baselines seemed to be obvious because APC can access the global information in the training phase.
Careful introduction may be necessary for the description of the proposed method. The detailed comments are described below.

**Questions:**

- 3.2: Careful introduction of the proposed method may be required. There have been many actor-critic algorithms and what and why the authors select the base algorithm may be necessary for understanding the formulation. In particular, before Eq. (1) is it better to add some explanations (with references)? And for minor points, T and V are undefined. In my understanding, the critic is not the paper’s original point.
- In my understanding, Loss_{prediction} is only the original point in this paper. Is it correct? Or is there any improved point? The methodological contribution other than predicting the global representations (i.e., Loss_{prediction}) was unclear to me.
- How about the ablation study about the GRF environment? I expected the results to be unstable, but some insights can be obtained.

---

> ### Author Response · Authors · 2023-11-18
>
> We wish to thank reviewer HFwe for carefully reviewing our paper and for the constructive comments! Please see below our responses. We use ***Q***, ***W***, ***A*** to denote question, weakness, and answer.
>
> >***Q1***
>
> ***A1***: Thanks for your suggestion, we'll add more detailed explanations and references in the latest version. In Eq.(1) $T$ denotes the batch size and $V_\phi(s)$ denotes the state value. The structure of the critic is mainly based on MAT (it has been mentioned in the paper).
>
> >***Q2***
>
> ***A2***: Yes, our main contribution is making the actor learn the global representations of the centralized critic so that the critic can provide more information to help with training. Although only adding a prediction loss seems like a small innovation, we believe that the idea of utilizing more information from the critic during training is concise and novel enough.
>
> >***Q3***
>
> ***A3***: Due to time constraints, we only conducted ablation experiments about the counterattack easy task in GRF, and the results are as follows:
>
> | APC | APC without GRU | APC without loss_pred |
> | -------- | -------- | -------- |
> | 0.938(0.044) | 0.854(0.097) | 0.896(0.064) |
>
> The results are still consistent with those in the paper.

---

> > ### Comment · Reviewer_HFwe · 2023-11-22
> > **Thanks for the response**
> >
> > Thanks for the response.
> > For A1 and A2, my direct concerns were clarified, but I expect an entire improvement of the clarity in this paper.
> > Regarding A3, thanks for the experiment. The result of the counterattack easy task may be according to your hypothesis.
> > Overall, at this stage, because of the unclarity and less novelty (relatively in this highly competitive conference), I will keep my rating.

---

### Meta-Review · Area_Chair_Ppcw · 2023-12-06

**Metareview:**

It reaches a consensus that the contribution and novelty of the paper in its current form are relatively limited, and some parts of the writing can be further improved. The experimental results may need to be strengthened, and the theoretical soundness of the proposed approaches may be worth further investigation. I suggest the authors incorporate the feedback and resubmit the paper to other upcoming ML venues.

**Justification For Why Not Higher Score:**

The paper still needs improvement to clear the bar of ICLR.

**Justification For Why Not Lower Score:**

N/A

---

### Decision · Program_Chairs · 2024-01-16

Reject